# Relation of MRI-Visible Perivascular Spaces and Other MRI Markers of Cerebral Small Vessel Disease

**DOI:** 10.3390/brainsci13091323

**Published:** 2023-09-14

**Authors:** Frances Rodriguez Lara, Arturo Ruben Toro, Adlin Pinheiro, Serkalem Demissie, Oluchi Ekenze, Oliver Martinez, Pedram Parva, Andreas Charidimou, Saptaparni Ghosh, Charles DeCarli, Sudha Seshadri, Mohamad Habes, Pauline Maillard, Jose Rafael Romero

**Affiliations:** 1Chobanian & Avedisian School of Medicine, Boston University, Boston, MA 02118, USA; frances2@bu.edu (F.R.L.); artoro@bu.edu (A.R.T.); 2Department of Biostatistics, School of Public Health, Boston University, Boston, MA 02118, USA; adlinp@bu.edu (A.P.); demissie@bu.edu (S.D.); 3Framingham Heart Study, National Heart Lung and Blood Institute, Framingham, MA 01702, USA; osekenze@bu.edu (O.E.); sapta@bu.edu (S.G.); seshadri@uthscsa.edu (S.S.); 4Graduate Medical Sciences, Chobanian & Avedisian School of Medicine, Boston University, Boston, MA 02118, USA; 5Department of Neurology, University of California Davis, Davis, CA 95817, USA; omartinez@ucdavis.edu (O.M.); cdecarli@ucdavis.edu (C.D.); pmaillard@ucdavis.edu (P.M.); 6Department of Radiology, Veterans Affairs Boston Healthcare System, Boston, MA 02118, USA; pedram.parvam@va.gov; 7Department of Radiology, Chobanian & Avedisian School of Medicine, Boston University, Boston, MA 02118, USA; 8Department of Neurology, Chobanian & Avedisian School of Medicine, Boston University, Boston, MA 02118, USA; antreas.charidimou@bmc.org; 9The Glenn Biggs Institute for Alzheimer’s and Neurodegenerative Diseases, University of Texas Health Sciences Center, San Antonio, TX 78229, USA; habes@uthscsa.edu

**Keywords:** MRI-visible perivascular spaces, cerebral small vessel disease, disease marker, glymphatic function

## Abstract

Perivascular spaces (PVS) visible on brain MRI signal cerebral small vessel disease (CSVD). The coexistence of PVS with other CSVD manifestations likely increases the risk of adverse neurological outcomes. We related PVS to other CSVD manifestations and brain volumes that are markers of vascular brain injury and neurodegeneration. Framingham Heart Study (FHS) participants with CSVD ratings on brain MRI were included. PVS were rated in the basal ganglia (BG) and centrum semiovale (CSO) into grades I–IV and a category reflecting high burden in single or mixed CSO-BG regions. We related PVS to covert brain infarcts (CBI), white matter hyperintensities (WMH), cerebral microbleeds (CMB), total brain, hippocampal, and cortical gray matter volumes using adjusted multivariable regression analyses. In 2454 participants (mean age 54 ± 12 years), we observed that higher PVS burden in both BG and CSO was related to CMB in lobar and deep brain regions and increased WMH. Greater CSO PVS burden was associated with decreased total cortical gray volumes. PVS are associated with ischemic markers of CSVD and neurodegeneration markers. Further studies should elucidate the causality between PVS and other CSVD manifestations.

## 1. Introduction

Cerebral small vessel disease (CSVD) comprises a group of diseases affecting the small arteries, arterioles, venules, and capillaries of the brain, and these diseases are strongly related to adverse neurological disorders [1]. Various brain magnetic resonance imaging (MRI) markers have emerged as indicators of CSVD and are used to quantify its severity, progression, and risk of adverse neurological outcomes such as incident stroke and dementia [2,3,4]. CSVD has major implications for public health: CSVD is the most common underlying type of cerebrovascular disease in patients with dementia, underlies 30% of ischemic strokes and is the most common cause of intracerebral hemorrhages.

MRI-visible perivascular spaces (PVS) have gained interest recently as markers of CSVD, potentially reflecting early stages of disease and also additional pathophysiological aspects of CSVD, namely glymphatic dysfunction. PVS are fluid-filled spaces surrounding the small perforating vessels in the brain [5], which become visible in high numbers representing ongoing vascular dysfunction. They may be implicated in impaired perivascular drainage of cerebral metabolites including β-amyloid and as such, represent disrupted cerebral homeostasis [6]. PVS can be best quantified using T2-weighted MRI sequences, allowing characterization of their burden across brain regions.

Study of CSVD using MRI markers has several aspects that require further definition. A key question arising in the study of brain MRI markers of vascular brain injury and clinical outcomes is if they are adding additional information beyond that provided by other well-characterized MRI markers. Thus, understanding the relation between PVS and other MRI markers of vascular brain injury and neurodegeneration is crucial to assess their independent contribution to neurological diseases and assess their role in the pathophysiology of stroke, dementia, and other neurological disorders which are frequently detectable in subclinical stages. However, the relations between neuroimaging markers are complex and incompletely characterized. Different CSVD markers frequently overlap and may be heterogeneous between patients.

Traditional markers of CSVD include white matter hyperintensities (WMH), covert brain infarcts (CBI), and cerebral microbleeds (CMB). WMH represent CSVD of ischemic nature, considered to reflect cerebral arteriole disease aspects such as stenosis, hypoperfusion and ischemia [7]. CMB, on the other hand, are considered a marker of hemorrhage-prone CSVD with topographic distributions reflecting differing underlying angiopathies, such as cerebral amyloid angiopathy (CAA) when located in lobar brain regions and hypertensive angiopathy when affecting solely deep regions or mixed brain regions in advanced cases. CMB may occur adjacent to perivascular spaces, potentially contributing to their enlargement with hemosiderin deposits from blood product extravasation [8,9]. CBI represent small ischemic cerebral infarctions and are frequently found incidentally. They are implicated in cognitive and motor decline, and are frequently progressive [10].

We aimed to characterize the relations of PVS to MRI markers of vascular brain injury and neurodegeneration in a large sample of community-dwelling individuals.

## 2. Materials and Methods

### 2.1. IRB Statement and Informed Consent

The study was conducted in accordance with the Declaration of Helsinki, and approved by the Institutional Review Board of Boston University School of Medicine for studies involving humans. Informed consent was obtained from all subjects.

### 2.2. Sample

The Framingham Heart Study is a prospective, multigenerational, population-based cohort study based in Framingham, MA, that began with the Original Cohort in 1948. Offspring of the original cohort and their spouses were then recruited as part of the Offspring Cohort in 1971; they were evaluated every 4 years. The Third Generation was recruited in 2002 and has been examined three times. For this study, we included Offspring and Third-Generation cohort participants with available brain MRI and ratings of CSVD features. Exclusion criteria included refusal or contraindication for MRI (pacemaker or other implantable devices, metallic foreign body, and claustrophobia), scans with significant artifacts precluding PVS assessment, scans without a corresponding clinic examination, and history of clinical stroke, dementia, and neurological conditions that could affect brain MRI measurements (such as head trauma, multiple sclerosis, brain tumor). Following these exclusions, the sample included 2452 scans from participants (649 offspring and 1803 third-generation participants).

### 2.3. Brain MRI

MRI acquisition and processing (brain volumes, white matter hyperintensity volume, and covert infarcts).

The MRI protocol in FHS has been described previously [11]. Briefly, participants were imaged by a variety of MRI machines varying in field strength from 1 to 3 Tesla. Two sequences were used: a 3-dimensional T1-weighted and 2- or 3-dimensional FLAIR imaging. All images were transferred to and processed by the University of California Davis Medical Center without knowledge of clinical information. Segmentation and quantification of brain volume measures were performed by automated procedures with quality control. Total cerebral cranial volume (TCV) was determined using a convolutional neural network method [12]. We obtained total brain volume using methods previously described [13]. To account for differences in head size, we used the total cranial-to-brain volume ratio (TCBV). Images were further segmented into 4 tissue types (gray matter, white matter, cerebrospinal fluid (CSF), and WMH volumes) using previously published methods [14,15,16]. Hippocampal analyses were performed using the atlas-based [17] diffeomorphic approach [18] with the minor modification of label refinement. Non-linear co-registration of images to the DKT atlas [19] enabled calculation of regional and global cortical gray matter volumes [18,20]. Regional volumes included the frontal, temporal, parietal, and occipital gray matter. Covert brain infarcts were identified according to standard protocols [7] as small ischemic cerebral lesions that are detected on MRI in the absence of clinical stroke events with excellent intra- and inter-rater reliability [21]. These ratings were categorized into two groups: any CBI (≥1) and small CBI (i.e., between 3 mm and 20 mm). Given that the relationship between PVS burden and WMH is not necessarily linear, we elected to model WMH volume as both a continuous and dichotomous variable (severe WMH). Severe WMH was defined as >1 age-specific standard deviation (SD) above the age-predicted value. We used this threshold for severe WMH burden in light of previous work relating decreased cognitive function to WMH volume 1 SD above the age-predicted value [22]. MRI measures that involved volumetric analyses were corrected for head size by calculating the ratio of these volumes over the total cranial volume and multiplied by 100 (percent TCV). The percent of WMH/TCV was log-transformed for normality.

### 2.4. MRI-Visible Perivascular Spaces (PVS) Rating

The MRI characteristics of PVS were based on criteria provided by the Standards for Reporting Vascular Changes on Neuroimaging (STRIVE) consortium [7]. Briefly, the PVS signal is similar to that of CSF on all sequences used, with a pattern of penetrating vessels, linear when parallel to the penetrating vessel or round/ovoid when perpendicular to the penetrating vessel), and a diameter smaller than 3 mm. Additional details of PVS assessments and reliability measures have been recently reported [23].

Scans were rated by trained investigators blinded to the subjects’ demographic and clinical information using T2-weighted axial or coronal MRI sequences [24]. We analyzed PVS in the centrum semiovale (CSO) and basal ganglia (BG) regions following a previously validated method based on individual PVS counts as seen in Figure 1 [24]. These counts were reflected in grades: Grade I (<10), Grade II (10–20), Grade III (20–40) and Grade IV (>40). We also studied results in mixed groups reflecting high PVS burden in deep and/or lobar brain regions as previously described [23]. We defined high PVS burden as grade III–IV in each region and then used this definition to create a categorical variable to describe high burden in the BG, CSO or both regions as follows: neither group, strictly BG group, strictly CSO group, or both BG and CSO group.

### 2.5. Cerebral Microbleeds

Cerebral microbleeds (CMB) were defined using standard criteria [25] as rounded or ovoid hypointense lesions on the T2*-GRE weighted sequence (Figure 2). The lesions measured 10 mm or less in diameter and were surrounded by brain parenchyma over at least half the circumference of the lesion. In the present study, CMB mimics were excluded.

Reliability measures for CSVD markers in the FHS have been previously reported, ranging from good to excellent. [23,26,27,28]

### 2.6. Covariates

Demographic and clinical characteristics were extracted from the exam cycle closest to MRI. Systolic (SBP) and diastolic (DBP) blood pressures (mmHg) were each taken as the average of the Framingham clinic physician’s two measurements. Hypertension status was evaluated using the JNC-7 criteria (SBP ≥ 140 mmHg or DBP ≥ 90 mmHg or use of antihypertensive medications). Antihypertensive medication use was noted for all cohorts. Current smoker status was a self-reported dichotomous variable. Diabetes was defined as fasting plasma glucose ≥126 mg/dL (≥7 mmol/L) for the offspring and third generation, or use of insulin or oral hypoglycemic medications for all cohorts.

### 2.7. Statistical Analysis

Descriptive statistics included mean (SD) or counts (%) for continuous and categorical clinical variables. The independent variables in our analysis were PVS burden (grades I–IV) in the BG and CSO separately and in the mixed regions, treated as categorical variables. The reference groups were grade I PVS in the BG and CSO regions and the subgroup without high burden PVS for the mixed categories.

Multivariable linear regression analyses were used to relate PVS in each brain region and the mixed grouping separately to total brain, WMH, hippocampal, total and regional cortical gray matter volumes. All volumes were log-transformed and then standardized to mean zero and variance one for analysis. Multivariable logistic regression was used to individually relate PVS to the presence of CBI, CMB, and severe WMH. The primary model adjusted for age, sex, FHS cohort, time interval between MRI acquisition and the clinic exam (Model 1). A second model additionally adjusted for vascular risk factors including hypertension, diabetes mellitus, and smoking (Model 2).

Due to the small sample of participants in subgroups of CMB topography across PVS categories, we collapsed PVS burden into two groups representing high burden (grades III and IV) versus low burden (grades I and II) in each region. The mixed region grouping in this case was a categorical variable describing the number of regions with high burden PVS: neither region, one region or both regions (BG and CSO).

Additional exploratory analyses assessed the relationship between PVS burden and total brain volume and regional cortical gray matter volumes in the frontal, parietal, temporal, and occipital lobes using multivariable linear regression. These values were also log-transformed and standardized and accounted for head size differences.

Given the large number of associations tested, we controlled the positive false discovery rate, using methods previously described [29], at 0.05 to assess whether our findings from the primary model retained significance in all but the exploratory analyses. Significance for the associations of the primary outcomes in Model 1 was defined by q-value < 0.05. A *p*-value < 0.05 (uncorrected) was considered statistically significant for the exploratory analyses and the estimates for Model 2. All analyses were performed using SAS version 9.4 (SAS Institute, Cary, NC, USA).

## 3. Results

### 3.1. Sample Characteristics

Baseline demographic characteristics for our sample are shown in Table 1. Our sample consisted of middle-aged participants well balanced for inclusion of men and women. Besides hypertension, participants had a low prevalence of vascular risk factors: 31% had hypertension, 8.5% were current smokers, 7.4% had diabetes mellitus, and approximately a fourth were using antihypertensive and lipid-lowering medications (24.30% and 23%, respectively).

### 3.2. Descriptive Statistics of Brain MRI Measures

The prevalence of CSVD markers in our sample was generally low as seen in Figure 3. CMB were seen in 5%, 7% had CBI, and 11% had severe WMH. Few participants had the highest burden of PVS in some subgroups. We observed that the proportion of participants with CMB, CBI and severe WMH increased as PVS burden increased regardless of the brain region. The opposite was also true; the proportion of participants without CMB, CBI or severe WMH decreased as PVS burden increased. As PVS burden increased, WMH volumes increased while hippocampal, total brain volume, and total and regional cortical gray matter volumes decreased regardless of the brain region where PVS was assessed (Table 2).

### 3.3. Multivariable Analysis

#### 3.3.1. MRI-Visible Perivascular Spaces and Covert Brain Infarcts (CBI) (Table 3A,B)

We observed significantly higher odds of any CBI in participants with grade III PVS compared to grade I in the BG and in the CSO in the primary model (Table 3A). These associations remained significant after adjustment for vascular risk factors (CSO OR: 1.99, 95% CI: 1.16, 3.43; BG OR: 5.27, 95% CI: 3.01, 9.22) (Table 3B). In addition, grade III PVS in the BG was associated with higher odds of small, any lobar, and any deep CBI. The associations were slightly stronger in Model 2.

In the mixed regions, the odds of CBI (any, small, and deep) were significantly greater in those with high PVS burden in both regions and in those with high burden in the BG region only, and after adjusting for vascular risk factors. The odds of lobar CBI were significantly higher only in participants with high burden in both regions.

#### 3.3.2. MRI-Visible PVS and White Matter Hyperintensity Volume (Table 4)

PVS burden was significantly associated with both WMH volume and severe WMH in both regions. Participants with either grade II, III, or IV PVS had higher odds of severe WMH and higher WMH volume compared to those with grade I PVS. Minimal attenuations were observed after adjusting for vascular risk factors, but the associations remained strong and statistically significant. Similar results were observed in the mixed regions where high burden, regardless of region, was associated with WMH volume and severe WMH.

#### 3.3.3. MRI-Visible Perivascular Spaces and Cerebral Microbleeds (Table 5A,B)

We did not observe any associations between PVS burden and the presence of any CMB or only lobar CMB, regardless of region (Table 5A,B). In exploratory analyses, we compared high burden (grade III–IV) vs. low burden (grade I–II) PVS in each region, and compared the number of regions with high burden PVS in the mixed CSO-BG regions (1 or 2 versus none). Those with high PVS burden in the CSO had significantly higher odds of lobar and deep CMBs after adjustment for vascular risk factors (OR: 4.74, 95% CI: 1.22, 18.40). In the primary model, high burden CSO PVS was associated with deep or mixed CMB (Table 5A), but this association was attenuated after adjustment for vascular risk factors (OR: 2.13, 95% CI: 0.95, 4.77) (Table 5B). In the mixed regions, participants with high PVS burden in both the BG and CSO had significantly higher odds of lobar and deep CMB in both models. However, these results need to be viewed with caution due to the small proportion of participants with CMB in some subgroups.

#### 3.3.4. MRI-Visible Perivascular Spaces and Hippocampal and Total Brain Volumes (Table 6)

Overall, there was an inverse relationship between higher PVS burden and TCBV and hippocampal volumes; as PVS burden increased in each region, TCBV and hippocampal volumes decreased. The relation of higher PVS burden and TCBV was significant at all grades of PVS burden in the BG and CSO regions, and after adjustment for vascular risk factors. We also observed a dose–effect relation where the effect size was stronger, i.e., lower total brain volumes as PVS burden increased. The dose-dependent, inverse relationship between PVS grade and TCBV was clearly illustrated by PVS in the BG. All associations showing the inverse relationship between PVS and TCBV were significant in Models 1 and 2.

These results were also observed in the mixed grouping where broadly, greater burden of PVS was associated with decreased hippocampal and TCBV. Of note, PVS burden in both the BG and CSO was not associated with a greater reduction in hippocampal or TCBV than PVS burden in the BG or CSO alone.

#### 3.3.5. MRI-Visible Perivascular Spaces and Total and Regional Cortical Gray Volumes (Table 7A,B)

Overall, we observed an inverse relation between higher PVS burden and cortical gray matter volumes in all regions examined (frontal, temporal, parietal, and occipital). We noted that PVS in the CSO was associated with lower cortical gray volumes in the frontal and temporal regions, while the same grades in the BG were associated with lower cortical gray volumes in the occipital and parietal regions; this relationship was statistically significant only in Model 1. For the most part, we noted a dose-dependent relationship where higher PVS grade was associated with lower cortical gray volumes. We continued to observe this dose dependence after controlling for vascular risk factors. In our mixed grouping of PVS burden, severe PVS burden strictly in the BG or CSO were associated with lower cortical gray volume in the occipital region; we noted a greater effect size when there was severe burden of PVS in both the BG and CSO. These relationships were attenuated when we controlled for vascular risk factors.

## 4. Discussion

We investigated the relations of PVS according to their topographic distribution in the brain with other markers of vascular brain injury, including markers of ischemic injury (CBI, WMH), hemorrhagic injury (CMB), and neurodegeneration (total brain and hippocampal volumes, and total and regional cortical gray matter volumes). We found that greater burden of PVS in both the CSO and BG was associated with the presence of any CBI, which remained significant after accounting for vascular risk factors. Greater burden of PVS was also associated with increasing WMH volume after controlling for vascular risk factors. High burden of PVS in the CSO and high burden in mixed CSO-BG regions were associated with lobar and deep CMB. Higher PVS burden in the BG and CSO were inversely related to total brain and hippocampal volumes, with a stronger relation with PVS in the CSO. We present a novel relation of PVS burden with total and regional cortical gray matter volumes, with results suggesting that high PVS burden is related to diffuse brain effects as reflected by a decrease in total cortical gray matter volumes, though some regional variations may exist.

These results expand prior reports by including a large community-based sample with younger participants than previously reported and addressing relations of PVS burden with neurodegeneration. We also summarize the relations of PVS with the most-studied CSVD markers, knowledge that is needed when considering these markers in relation to adverse neurological outcomes. A prior study found a relationship between increased PVS burden and increased WMH volume and the presence of lobar CMB [30]. In a population-based study in a Japanese cohort, BG PVS were associated with deep CMB, while lobar PVS were associated with greater burden of lobar CMB [31]. This study also showed a relationship between PVS in both the BG and CSO with increased volume of WMH, which is consistent with our findings. The Three-City Dijon MRI study also showed a relationship with greater burden of PVS and WMH [32]. PVS were also shown in the Reykjavik-AGES study to be associated with increased WMH and greater burden of incidental subcortical brain infarcts [33].

We found that increasing burden of MRI-visible PVS in both the BG and the CSO were strongly associated with greater volume of WMH and higher prevalence of CBI. The relations between these CSVD markers suggest shared processes involving vascular dysfunction as possible mechanisms. For instance, impaired function of the blood–brain barrier, endothelial dysfunction, and hypoperfusion have been related to PVS, WMH and CBI [34,35]. Ischemic injury may also be part of a vicious cycle where initial triggers lead to ongoing vascular injury that in turn promotes further ischemic injury. The development of vascular insufficiency could drive the enlargement of the perivascular space [36]. This vascular insufficiency may also play a role in the chronically hypo-perfused environments that have been shown surrounding white matter lesions [37]. Other shared mechanisms that could participate in the development and progression of these markers include oxidative stress [38,39], persistent inflammation [40,41] and endothelial dysfunction, leading to parenchymal damage, astrocytic and neuronal injury, which may alter the size of the perivascular spaces [42].

It is important to consider that while PVS in both brain regions and ischemic changes were found concurrently, the prior literature suggests that in some cohorts (i.e., patients with intracerebral hemorrhage), PVS in the BG and CSO may be associated with different sets of risk factors and different etiologies: PVS in the CSO have been associated with CAA and greater concentrations of CSF markers of Alzheimer’s Disease [43] whereas PVS in the BG is associated with hypertensive arteriopathy [44,45].

Similar mechanisms to the ones mentioned with ischemic markers may play a role in the relations of PVS and CMB. In addition, prior studies link increased arterial stiffness with BG CMB and PVS in both the BG and CSO [46], which could reflect the effects of hypertensive arteriopathy.

We found that increasing burden of PVS, particularly in the CSO, was inversely related to hippocampal and total brain volumes. A previous study looking at the relationship between BG PVS burden and brain atrophy showed that brain atrophy was an independent risk factor for severe BG-PVS in lacunar stroke patients [47]. Another population- based study showed that severity of PVS in healthy, elderly individuals was not associated with brain atrophy [32]. Notably, in this study, we demonstrated that all grades of PVS in either the BG or CSO were associated with significantly decreased TCBV with a dose-dependent effect, indicating global brain atrophy.

Similarly, there was an inverse relation between PVS burden and cortical gray matter volumes both globally, and with regional variations. We found that PVS burden in the BG may correspond with greater atrophy of the parietal and occipital lobes while PVS burden in the CSO corresponds with atrophy of the temporal and frontal lobes. These findings are in agreement with a prior study that showed that increased PVS burden in the BG was broadly associated with decreases in cortical gray volume and also in specific portions of the frontal, temporal, parietal, and occipital brain regions [48]. These markers are suggestive of neurodegenerative processes, supporting a possible role for PVS as correlates of neurodegeneration [49].

In addition to their role as markers of stroke risk, PVS have been related to long-term cognitive decline [50]. Greater burden of PVS at time of death was related to lower levels of both semantic memory and visuospatial abilities [36]. Beyond its association with the magnitude of cognitive decline, individuals with greater PVS burden show greater rates of cognitive decline and incidence of dementia [51]. We have recently showed increased risk of incident dementia among participants with greater PVS burden [52].

The strong relationship we found between PVS with other CSVD and neurodegeneration markers suggest that research evaluating PVS in relation to clinical outcomes should consider assessing the independent relation from other CSVD and atrophy markers. However, it is important to note that although strong, such relationship may not apply to every individual and that, for some individuals, different markers may play a more predominant role.

Strengths of our study included a large sample of community-dwelling participants, including both men and women across a spectrum of ages. The presence of different forms of CSVD on MRI was noted, which was rated by experienced raters blinded to clinical context, using methods previously validated, with good to excellent reproducibility. A novelty of these findings is that it supports PVS as a neurodegenerative marker given the relationship between increased PVS burden and decreased total brain, total cortical gray volume globally and in separate brain regions. This is in agreement with prior works and contributes new information by exploring how PVS burden in the BG and CSO contribute to brain atrophy to varying extents depending on the brain region assessed. Our study also has several limitations to consider. An important limitation is that the small sample with highest grade PVS (grade IV) and markers such as WMHs and CBI were small, thus limiting meaningful statistical analyses. However, from our data, the trend is maintained that greater PVS burden in either the BG or the CSO is found together with a greater burden of CSVD markers. The low prevalence of CMB in our sample limited the analyses relating PVS burden to CMB presence; thus, those results should be viewed with caution given the small sample size of participants with CMB and the presence of other studies suggesting that there is no such relationship [53]. Our study included primarily participants of White race, thus limiting the generalization of results to other racial groups.

Future directions in the study of PVS include the development of automated methods for accurate volumetric characterization of the glymphatic system represented by visible PVS. Such methods are in development by several groups with promising early results, which include allowing use of low-resolution scans with significant background noise [54]. The main advantage of such approach is the identification of PVS distribution in a 3D manner. Studies applying such methods in various contexts are needed, including population-based settings and clinical samples. In addition to technical development, further studies are needed to understand the longitudinal occurrence of various CSVD manifestations, characterize early versus advanced markers, and their clinical application for the prevention of stroke and dementia.

## 5. Conclusions

In conclusion, we found increased burden of PVS to be related to CSVD markers of ischemic, hemorrhagic, and neurodegenerative injury. Particularly, a greater burden of PVS in both the CSO and BG was associated with the presence of any CBI and increasing WMH, even after accounting for vascular risk factors. These findings support commonality in the process relating to vascular dysfunction as possible mechanisms. High burden of PVS in the CSO and high burden in mixed CSO-BG regions were associated with lobar and deep CMB. Further work is needed to better understand the vulnerabilities of cerebral vasculature, and mechanisms of shared and differing pathophysiology among CSVD markers. For example, given that PVS is associated strongly with ischemic CSVD markers and associated tenuously with hemorrhagic CSVD markers, pathological works could help clarify where these forms of CSVD diverge mechanistically. Additionally, little is understood regarding the pathophysiology of neurodegeneration and cerebral atrophy that has been noted to accompany CSVD. A better understanding of the cascade of injury represented by PVS and other CSVD markers can help us know how and when to intervene therapeutically.

## Figures and Tables

**Figure 1 brainsci-13-01323-f001:**
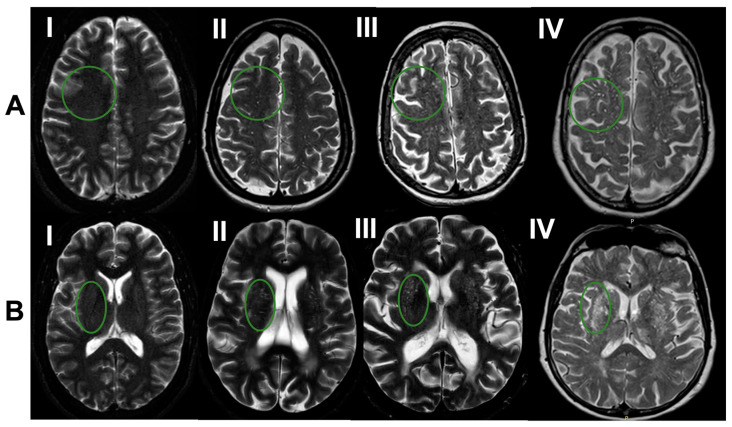
MRI-visible perivascular spaces (PVS) ratings by region: (**A**) centrum semiovale and (**B**) basal ganglia. PVS in each region were graded independently using counts to assign severity: grade I (<10), grade II (10–20), grade III (20–40) and grade IV (>40). The green circle represents a region enclosing PVS.

**Figure 2 brainsci-13-01323-f002:**
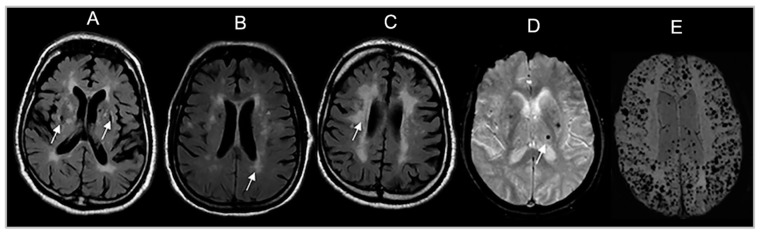
Brain MRI markers of cerebral small vessel disease: (**A**) Covert brain infarct (CBI). (**B**) White matter hyperintensities (WMH). (**C**). Severe WMH. (**D**). Single cerebral microbleed (CMB). (**E**). Innumerable CMB. White arrows indicate lesions of interest.

**Figure 3 brainsci-13-01323-f003:**
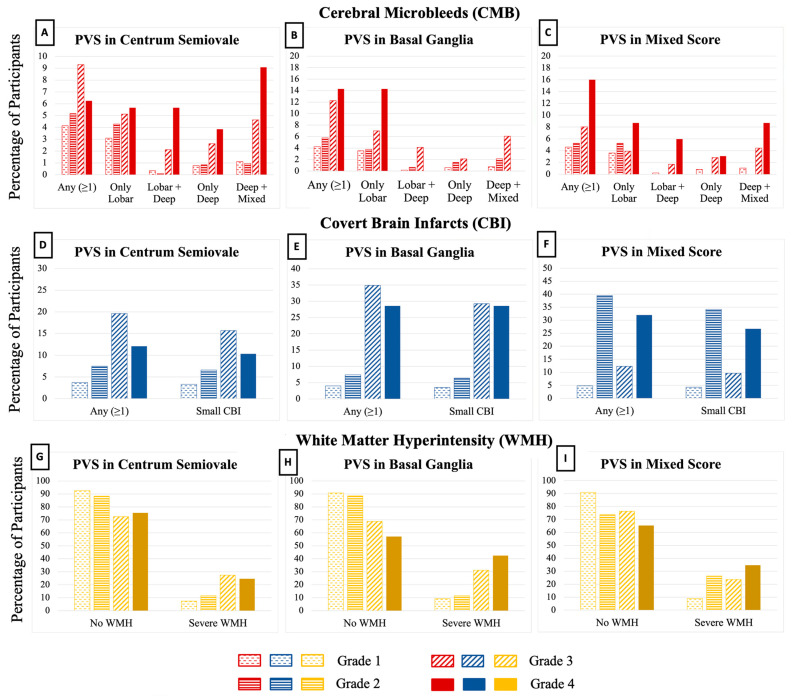
Burden and topography of MRI-visible Perivascular Spaces (PVS) with corresponding prevalence of cerebral microbleeds, covert brain infarcts and severe white matter hyperintensities. PVS were graded independently in the basal ganglia (**B**,**E**,**H**) and centrum semiovale (**A**,**D**,**G**) using counts to assign severity: grade I (<10), grade II (10–20), grade III (20–40) and grade IV (>40). Mixed group categories of high burden (**C**,**F**,**I**) are none, only basal ganglia, only centrum semiovale, and both regions.

**Table 1 brainsci-13-01323-t001:** Sample characteristics.

Clinical Characteristics	All N = 2452
Male, n (%)	1178 (48)
Age at MRI, years, mean (SD)	54.1 (12.1)
Age at clinic exam, years, mean (SD)	52.1 (12.4)
Time interval between MRI and clinic exam, years, mean (SD)	1.4 (1.0)
FHS Cohort, n (%)	
Offspring	649 (26)
Third Generation	1803 (74)
**Vascular risk factors**	
Systolic blood pressure, mm Hg, mean (SD)	119.6 (15.6)
Diastolic blood pressure, mm Hg, mean (SD)	74.1 (9.6)
Hypertension ^a^, n (%)	758 (31)
Current smoker, n (%)	208 (8)
Diabetes mellitus, n (%)	180 (7)
Body mass index, kg/m^2^, mean (SD)	28.0 (5.5)
APOE-ɛ4 (N = 2348), n (%)	541 (23)
Lipid-lowering medication use, n (%)	564 (23)
Antihypertensive use, n (%)	594 (24)
**MRI-visible Perivascular Spaces**	
Centrum Semiovale (CSO), n (%)	
Grade I	1205 (49)
Grade II	986 (40)
Grade III	204 (8)
Grade IV	57 (2)
Basal Ganglia (BG), n (%)	
Grade I	1372 (56)
Grade II	967 (39)
Grade III	106 (4)
Grade IV	7 (1)
Mixed region high PVS burden ^b^,n (%)	
None	2153 (88)
Basal Ganglia Only	38 (2)
Centrum Semiovale Only	186 (8)
Both	75 (3)
**Cerebral Microbleeds, n (%)**	
Any (≥1)	128 (5)
Only Lobar (N = 2416)	92 (4)
Lobar and Deep (N = 2336)	12 (1)
Only Deep (N = 2348)	24 (1)
Deep or Mixed (N = 2360)	36 (2)
**Covert Brain Infarcts**	
Any (≥1)	165 (7)
Small	142 (6)
**White Matter Hyperintensity**	
Severe White Matter Hyperintensity, n (%)	274 (11)
White Matter Hyperintensity Volume ^c^, mean (SD)	0.03 (0.85)
**Other Volumes**	
Hippocampal volume cm^3^, mean (SD)	6.81 (0.74)
Cortical gray volume cm^3^, mean (SD)	501.92 (51.22)
Total brain volume cm^3^, mean (SD)	1006.66 (107.96)
Frontal cortical gray volume cm^3^, mean (SD)	194.02 (21.27)
Temporal cortical gray volume cm^3^, mean (SD)	133.88 (13.96)
Parietal cortical gray volume cm^3^, mean (SD)	107.00 (11.32)
Occipital cortical gray volume cm^3^, mean (SD)	67.02 (8.83)

FHS = Framingham Heart Study; MRI = magnetic resonance imaging; PVS = perivascular spaces; SD = standard deviation. ^a^ Hypertension is defined as SBP ≥ 140 mmHg or DBP ≥ 90 mm Hg or current anti-hypertensive use. ^b^ High PVS burden is defined as grades III–IV in the respective region. ^c^ Log-transformed then standardized to mean 0 and variance 1.

**Table 2 brainsci-13-01323-t002:** (**A**). Prevalence of CSVD markers and brain volumes stratified by MRI-visible perivascular space (PVS) burden and brain topography. (**B**). Prevalence of CSVD markers and brain volumes stratified by regions of high-MRI-visible perivascular spaces (PVS) burden.

(A)									
		MRI-Visible Perivascular Spaces ^a^ N = 2452							
		Centrum Semiovale					Basal Ganglia		
		I	II	III	IV	I	II	III	IV
**Cerebral** **Microbleeds (CMB)**	None N (%)	1155 (96)	935 (95)	185 (91)	49 (86)	1314 (96)	911 (94)	93 (88)	6 (86)
	Any (≥1), N (%)	50 (4)	51 (5)	19 (9)	8 (14)	58 (4)	56 (6)	13 (12)	1 (14)
	Only Lobar, N (%)	37 (3)	42 (4)	10 (5)	3 (6)	48 (4)	36 (4)	7 (7)	1 (14)
	Lobar and Deep, N (%)	4 (0)	1 (0)	4 (2)	3 (6)	2 (0)	6 (1)	4 (4)	0 (0)
	Only Deep, N (%)	9 (1)	8 (1)	5 (3)	2 (4)	8 (1)	14 (2)	2 (2)	0 (0)
	Deep or Mixed, N (%)	13 (1)	9 (1)	9 (5)	5 (9)	10 (1)	20 (2)	6 (6)	0 (0)
**Covert Brain Infarcts** **(CBI)**	None N (%)	1160 (96)	912 (92)	164 (80)	51 (89)	1317 (96)	896 (93)	69 (65)	5 (71)
	Any (≥1), N (%)	45 (4)	74 (8)	40 (20)	6 (11)	55 (4)	71 7)	37 (35)	2 (29)
	Small CBI, N (%)	40 (3)	65 (7)	32 (16)	5 (9)	48 (3)	6 (6)	31 (29)	2 (29)
**White** **Matter Hyperintensities (WMH)**	No Severe WMH, N (%)	1116 (93)	871 (88)	148 (73)	43 (75)	1245 (91)	856 (89)	73 (69)	4 (57)
	Severe WMH, N (%)	89 (7)	115 (12)	56 (27)	14 (25)	127 (9)	111 (11)	33 (31)	3 (43)
	WMH Volume ^‡^, Mean (SD)	−0.12 (0.82)	0.07 (0.83)	0.51 (0.89)	0.55 (0.85)	−0.02 (0.81)	0.03 (0.87)	0.52 (0.87)	1.16 (1.02)
**Other** **Volumes**	Hippocampal Volume cm^3^, Mean (SD)	6.87 (0.73)	6.79 (0.71)	6.64 (0.77)	6.49 (0.90)	6.84 (0.72)	6.81 (0.73)	6.50 (0.86)	6.36 (0.67)
	Cortical Gray Volume cm^3^, Mean (SD)	507.04 (50.55)	500.76 (50.79)	486.03 (49.86)	470.74 (54.78)	504.63 (49.79)	501.55 (52.10)	473.61 (51.68)	452.77 (53.35)
	Total brain volume cm^3^, mean (SD)	1015.76 (106.71)	1005.3 (106.9)	976.24 (108.19)	946.67 (114.06)	1011.45 (105.53)	1006.82 (109.26)	949.19 (110.19)	915.55 (93.39)
	Frontal cortical gray volume cm^3^, mean (SD)	196.9 (20.92)	192.93 (20.93)	186.45 (20.9)	179.13 (21.52)	195.2 (20.4)	193.89 (21.84)	181.2 (22.27)	174.29 (23.44)
	Temporal cortical gray volume cm^3^, mean (SD)	134.81 (14.02)	133.7 (13.77)	130.72 (13.64)	128.52 (14.71)	134.26 (13.73)	134.01 (14.23)	128.48 (13.34)	121.83 (12.51)
	Parietal cortical gray volume cm^3^, mean (SD)	107.79 (11.05)	106.98 (11.28)	104.06 (11.7)	101.29 (13.14)	107.63 (11.02)	106.84 (11.49)	101 (11.63)	98.34 (13.5)
	Occipital cortical gray volume cm^3^, mean (SD)	67.54 (8.66)	67.15 (8.99)	64.81 (8.33)	61.8 (8.69)	67.53 (8.77)	66.81 (8.82)	62.93 (8.5)	58.3 (7.69)
**(B)**									
		**High-Burden PVS Regions ^a^** **N = 2452**							
		**None** **N = 2153**		**Only Basal Ganglia** **N = 38**		**Only Centrum Semiovale** **N = 186**		**Both** **N =75**	
**Cerebral** **Microbleeds (CMB)**	None N (%)	2054 (95)		36 (95)		171 (92)		63(84)	
	Any (≥1) N (%)	99 (5)		2 (5)		15 (8)		12 (16)	
	Only Lobar N (%)	77 (4)		2 (5)		7 (4)		6 (9)	
	Lobar and Deep N (%)	5 (0)		0 (0)		3 (2)		4 (6)	
	Only Deep N (%)	17 (1)		0 (0)		5 (3)		2 (3)	
	Deep or Mixed N (%)	22 (1)		0 (0)		8 (4)		6 (9)	
**Covert Brain Infarcts** **(CBI)**	None N (%)	2049 (95)		23 (61)		164(88)		51(68)	
	Any (≥1) N (%)	104 (5)		15 (39)		22 (12)		24 (32)	
	Small CBI N (%)	92 (4)		13 (34)		17 (9)		20 (27)	
**White Matter Hyperintensity** **(WMH)**	No Severe WMH, N(%)	1959 (91)		28 (74)		142(76)		49(65)	
	Severe WMH N (%)	194 (9)		10 (26)		44 (24)		26 (35)	
	WMH Volume ^‡^, Mean (SD)	−0.04 (0.82)		0.36 (1.02)		0.46 (0.91)		0.66 (0.80)	
**Other** **Volumes**	Hippocampal cm^3^, Mean (SD)	6.84 (0.72)		6.65 (0.72)		6.69 (0.75)		6.41 (0.90)	
	Cortical Gray cm^3^, Mean (SD)	504.52 (50.70)		487.04 (50.79)		489.88 (49.71)		464.86 (51.00)	

(**A**) FHS = Framingham Heart Study; MRI = magnetic resonance imaging; PVS = perivascular spaces; SD = standard deviation ^a^ PVS were graded independently in the basal ganglia and centrum semiovale using counts to assign severity: grade I (<10), grade II (10–20), grade III (20–40) and grade IV (>40). ^‡^ Log-transformed then standardized to mean 0 and variance 1. (**B**) MRI = magnetic resonance imaging; PVS = perivascular spaces; SD = standard deviation; ^a^ PVS were graded independently in the basal ganglia and centrum semiovale using counts to assign severity: grade I (<10), grade II (10–20), grade III (20–40) and grade IV (>40). High PVS burden was defined as grade III–IV in each region. ^‡^ Log-transformed then standardized to mean 0 and variance 1.

**Table 3 brainsci-13-01323-t003:** (**A**). Multivariable analysis of the relation between perivascular space burden by grade ^a^ and brain topography and covert brain infarcts (Model 1 ^‡^). (**B**). Multivariable analysis of the relation of perivascular space burden by grade ^a^ and brain topography and covert brain infarcts (Model 2 ^‡^).

(A)						
MRI-Visible Perivascular Spaces (PVS) Grading ^a^		Covert Brain Infarcts (CBI) N = 2452				
		Model 1 ^‡^ OR (95% CI)				
		Any CBI	Small CBI	Any Lobar	Any Deep	
**Centrum Semiovale**	I	Ref	Ref	Ref	Ref	
	II	1.28 (0.85, 1.92)	1.27 (0.82, 1.95)	0.84 (0.45, 1.57)	1.85 * (1.03, 3.33)	
	III	2.08 * (1.22, 3.56)	1.81 (1.02, 3.22)	1.62 (0.73, 3.57)	2.08 (0.98, 4.41)	
	IV	0.90 (0.34, 2.36)	0.85 (0.30, 2.41)	0.61 (0.13, 2.94)	0.65 (0.14, 3.07)	
**Basal** **Ganglia**	I	Ref	Ref	Ref	Ref	
	II	1.44 (0.98, 2.10)	1.42 (0.95, 2.12)	1.20 (0.67, 2.13)	1.81 * (1.06, 3.11)	
	III	4.96 ** (2.86, 8.60)	4.52 ** (2.52, 8.09)	2.71 * (1.18, 6.25)	6.17 ** (3.04, 12.52)	
	IV	2.76 (0.49, 15.64)	3.38 (0.59, 19.32)	2.84 (0.30, 26.82)	2.79 (0.30, 25.66)	
**Mixed** **Region High PVS Burden**	None	Ref	Ref	Ref	Ref	
	Only BG	5.52 ** (2.63, 11.57)	5.00 ** (2.33, 10.76)	2.62 (0.82, 8.35)	7.20 ** (3.20, 16.19)	
	Only CSO	1.28 (0.75, 2.18)	1.10 (0.61, 1.99)	1.36 (0.61, 3.04)	1.01 (0.48, 2.11)	
	Both	3.46 ** (1.90, 6.30)	3.06 ** (1.63, 5.77)	2.68 * (1.12, 6.42)	2.66 * (1.23, 5.76)	
**(B)**						
**MRI-Visible** **Perivascular Spaces (PVS) Grading ^a^**		**Covert Brain Infarcts (CBI)** **N = 2452**				
		**Model 2 ^‡^ OR (95% CI)**				
		**Any CBI**	**Small CBI**	**Any Lobar**	**Any Deep**	
**Centrum Semiovale (CSO)**	I	Ref	Ref	Ref	Ref	
	II	1.28 (0.85, 1.93)	1.27 (0.82, 1.96)	0.83 (0.44, 1.57)	1.85 * (1.02, 3.33)	
	III	1.99 * (1.16, 3.43)	1.71 (0.96, 3.07)	1.62 (0.73, 3.61)	1.90 (0.89, 4.07)	
	IV	0.95 (0.36, 2.52)	0.90 (0.31, 2.57)	0.66 (0.14, 3.16)	0.66 (0.14, 3.13)	
**Basal** **Ganglia (BG)**	I	Ref	Ref	Ref	Ref	
	II	1.43 (0.98, 2.10)	1.42 (0.94, 2.12)	1.20 (0.68, 2.15)	1.79 * (1.04, 3.08)	
	III	5.27 ** (3.01, 9.22)	4.74 ** (2.63, 8.55)	2.81 * (1.21, 6.54)	6.58 ** (3.22, 13.47)	
	IV	2.79 (0.49, 15.95)	3.40 (0.59, 19.59)	2.74 (0.28, 26.50)	3.01 (0.33, 27.73)	
**Mixed** **Region High PVS Burden**	None	Ref	Ref	Ref	Ref	
	Only BG	5.89 ** (2.79, 12.44)	5.24 ** (2.42, 11.33)	2.83 (0.88, 9.07)	7.49 ** (3.31, 16.96)	
	Only CSO	1.22 (0.71, 2.09)	1.03 (0.56, 1.88)	1.42 (0.64, 3.18)	0.86 (0.40, 1.87)	
	Both	3.57 ** (1.94, 6.56)	3.11 ** (1.64, 5.91)	2.73 * (1.13, 6.64)	2.73 * (1.25, 5.96)	

(**A**) BG = basal ganglia; CI = confidence interval; CSO = centrum semiovale; FHS = Framingham Heart Study; MRI = magnetic resonance imaging; OR = odds ratio; PVS = perivascular spaces. ^a^ PVS were graded independently in the basal ganglia and centrum semiovale using counts to assign severity: grade I (<10), grade II (10–20), grade III (20–40) and grade IV (>40). High PVS burden was defined as grade III–IV in each region. ^‡^ Model 1 adjusts for age, sex, FHS cohort, and time interval between MRI acquisition and the clinic exam. * q-values < 0.05; ** q-values < 0.001. (**B**) BG = basal ganglia; CI = confidence interval; CSO = centrum semiovale; FHS = Framingham Heart Study; MRI = magnetic resonance imaging; OR = odds ratio; PVS = perivascular spaces; ^a^ PVS were graded independently in the basal ganglia and centrum semiovale using counts to assign severity: grade I (<10), grade II (10–20), grade III (20–40) and grade IV (>40). High PVS burden was defined as grade III–IV in each region. ^‡^ Model 2 adjusts for age, sex, FHS cohort, time interval between MRI acquisition and the clinic exam, hypertension, diabetes mellitus, and smoking status; * *p*-values < 0.05; ** *p*-values < 0.001.

**Table 4 brainsci-13-01323-t004:** Multivariable analysis of the relation of MRI-visible perivascular spaces by grade ^a^ and brain region, and white matter hyperintensity burden and volume (Models 1 and 2 ^‡^).

MRI-Visible Perivascular Spaces (PVS) Grading ^a^		White Matter Hyperintensity (WMH) N = 2452			
		Model 1 ^‡^		Model 2 ^‡^	
		Severe WMH OR (95% CI)	WMH Volume β (95% CI)	Severe WMH OR (95% CI)	WMH Volume β (95% CI)
**Centrum Semiovale (CSO)**	I	Ref	Ref	Ref	Ref
	II	2.01 ** (1.48, 2.73)	0.28 ** (0.21, 0.35)	1.96 ** (1.44, 2.67)	0.27 ** (0.20, 0.34)
	III	7.32 ** (4.66, 11.52)	0.82 ** (0.68, 0.95)	7.17 ** (4.53, 11.33)	0.81 ** (0.67, 0.94)
	IV	6.74 ** (3.32, 13.68)	0.89 ** (0.67, 1.12)	5.58 ** (2.65, 11.75)	0.85 ** (0.61, 1.08)
**Basal** **Ganglia (BG)**	I	Ref	Ref	Ref	Ref
	II	1.39 * (1.06, 1.83)	0.10 * (0.03, 0.17)	1.35 * (1.02, 1.78)	0.09 * (0.02, 0.16)
	III	5.58 ** (3.35, 9.29)	0.65 ** (0.48, 0.82)	5.46 ** (3.25, 9.18)	0.65 ** (0.47, 0.83)
	IV	9.41 * (1.99, 44.41)	1.29 ** (0.67, 1.91)	9.05 * (1.92, 42.81)	1.28 ** (0.66, 1.9)
**Mixed** **Region High PVS Burden**	None	Ref	Ref	Ref	Ref
	Only BG	5.18 ** (2.38, 11.29)	0.53 ** (0.26, 0.8)	5.30 ** (2.42, 11.60)	0.55 ** (0.28, 0.82)
	Only CSO	4.13 ** (2.71, 6.28)	0.60 ** (0.47, 0.73)	3.97 ** (2.59, 6.08)	0.58 ** (0.45, 0.71)
	Both	8.82 ** (4.96, 15.69)	0.89 ** (0.69, 1.09)	8.55 ** (4.75, 15.37)	0.88 ** (0.68, 1.08)

BG = basal ganglia; CI = confidence interval; CSO = centrum semiovale; FHS = Framingham Heart Study; MRI = magnetic resonance imaging; OR = odds ratio; PVS = perivascular spaces; ^a^ PVS were graded independently in the basal ganglia and centrum semiovale using counts to assign severity: grade I (<10), grade II (10–20), grade III (20–40) and grade IV (>40). High PVS burden was defined as grade III–IV in each region.; ^‡^ Model 1 adjusts for age, sex, FHS cohort, and time interval between MRI acquisition and the clinic exam. Model 2 additionally adjusts for diabetes mellitus, hypertension, and smoking status; * *p*-values or q-values < 0.05; ** *p*-values or q-values < 0.001.

**Table 5 brainsci-13-01323-t005:** (**A**). Multivariable analysis of the relation of MRI-visible perivascular spaces by grade ^a^ and brain region and cerebral microbleed burden (Model 1 ^‡^). (**B**). Multivariable analysis of the relation of MRI-visible perivascular spaces by grade ^a^ and brain region, and cerebral microbleed burden (Model 2 ^‡^).

(A)								
MRI-Visible Perivascular Spaces (PVS) Grading ^a^				Cerebral Microbleeds (CMB)				
				Model 1 ^‡^ OR (95% CI)				
				Any CMB N = 2452	Only Lobar N = 2416	Lobar and Deep N = 2336	Only Deep N = 2348	Deep or Mixed N = 2360
**Centrum Semiovale (CSO)**		I		Ref	Ref	Ref	Ref	Ref
		II		0.86 (0.56, 1.31)	1.01 (0.63, 1.63)			
		III		0.99 (0.52, 1.86)	0.80 (0.36, 1.79)	4.89 * (1.28, 18.65)	1.62 (0.59, 4.44)	2.40 * (1.09, 5.29)
		IV		1.34 (0.55, 3.23)	0.79 (0.22, 2.84)			
**Basal** **Ganglia (BG)**		I		Ref	Ref	Ref	Ref	Ref
		II		1.06 (0.71, 1.56)	0.86 (0.54, 1.36)			
		III		1.25 (0.62, 2.54)	0.93 (0.38, 2.28)	3.28 (0.80, 13.41)	0.70 (0.15, 3.29)	1.47 (0.54, 3.95)
		IV		1.20 (0.14, 10.60)	1.73 (0.19, 15.48)			
**Mixed** **Region High PVS Burden**		None		Ref	Ref	Ref	Ref	Ref
		Only BG		0.52 (0.12, 2.28)	0.74 (0.17, 3.28)	2.52 (0.5, 12.73)	1.29 (0.42, 3.94)	1.57 (0.62, 3.93)
		Only CSO		0.97 (0.52, 1.78)	0.64 (0.28, 1.46)			
		Both		1.53 (0.74, 3.15)	1.08 (0.42, 2.78)	8.27 * (1.60, 42.63)	1.25 (0.25, 6.30)	2.86 (0.98, 8.38)
**(B)**								
**MRI-visible** **Perivascular Spaces (PVS) Grading ^a^**			**Cerebral Microbleeds (CMB)**					
			**Model 2 ^‡^ OR (95% CI)**					
			**Any CMB** **N = 2452**		**Only Lobar** **N = 2416**	**Lobar and Deep** **N = 2336**	**Only Deep** **N = 2348**	**Deep or Mixed** **N = 2360**
**Centrum Semiovale (CSO)**	I		Ref		Ref	Ref	Ref	Ref
	II		0.83 (0.54, 1.29)		1.01 (0.62, 1.64)			
	III		0.96 (0.51, 1.84)		0.82 (0.36, 1.84)	4.80 * (1.24, 18.62)	1.33 (0.46, 3.84)	2.14 (0.95, 4.78)
	IV		1.18 (0.47, 2.99)		0.84 (0.23, 3.07)			
**Basal Ganglia (BG)**	I		Ref		Ref	Ref	Ref	Ref
	II		1.06 (0.71, 1.57)		0.89 (0.56, 1.41)			
	III		1.26 (0.62, 2.57)		0.94 (0.38, 2.32)	3.20 (0.80, 12.81)	0.76 (0.16, 3.71)	1.57 (0.58, 4.24)
	IV		1.12 (0.13, 9.92)		1.62 (0.18, 14.57)			
**Mixed** **Region High PVS Burden**	None		Ref		Ref	Ref		Ref
	Only BG		0.54 (0.12, 2.36)		0.76 (0.17, 3.35)	2.50 (0.49, 12.72)	1.04 (0.31, 3.43)	1.35 (0.52, 3.52)
	Only CSO		0.93 (0.5, 1.75)		0.67 (0.29, 1.55)			
	Both		1.50 (0.72, 3.09)		1.07 (0.42, 2.78)	7.74 * (1.50, 39.93)	1.21 (0.24, 6.09)	2.73 (0.93, 8.01)

(**A**) BG = basal ganglia; CI = confidence interval; CSO = centrum semiovale; FHS = Framingham Heart Study; MRI = magnetic resonance imaging; OR = odds ratio; PVS = perivascular spaces; ^a^ PVS were graded independently in the basal ganglia and centrum semiovale using counts to assign severity: grade I (<10), grade II (10–20), grade III (20–40) and grade IV (>40). High PVS burden was defined as grade III–IV in each region. ^‡^ Model 1 adjusts for age, sex, FHS cohort, time interval between MRI acquisition and the clinic exam; * q-values < 0.05. (**B**) BG = basal ganglia; CI = confidence interval; CSO = centrum semiovale; FHS = Framingham Heart Study; MRI = magnetic resonance imaging; OR = odds ratio; PVS = perivascular spaces; ^a^ PVS were graded independently in the basal ganglia and centrum semiovale using counts to assign severity: grade I (<10), grade II (10–20), grade III (20–40) and grade IV (>40). High PVS burden was defined as grade III–IV in each region. ^‡^ Model 2 adjusts for age, sex, FHS cohort, time interval between MRI acquisition and the clinic exam, diabetes mellitus, hypertension, and smoking status; * *p*-values < 0.05.

**Table 6 brainsci-13-01323-t006:** Multivariable analysis of the relation between MRI-visible perivascular spaces by grade and brain region and total cranial-to-brain volume ratio and hippocampal volumes.

MRI-Visible Perivascular Spaces (PVS) Grading ^a^		Volumes N = 2452			
		Model 1 ^‡^		Model 2 ^‡^	
		Total Cranial to Brain Volume Ratio β (95% CI)	Hippocampal β (95% CI)	Total Cranial to Brain Volume Ratio β (95% CI)	Hippocampal β (95% CI)
**Centrum Semiovale (CSO)**	I	Ref	Ref	Ref	Ref
	II	−0.06 * (−0.11, −0.01)	−0.09 * (−0.17, −0.02)	−0.04 (−0.09, 0.01)	−0.09 * (−0.17, −0.01)
	III	−0.24 ** (−0.33, −0.14)	−0.23 * (−0.38, −0.08)	−0.21 ** (−0.31, −0.12)	−0.24 * (−0.38, −0.09)
	IV	−0.22 * (−0.38, −0.06)	−0.21 (−0.46, 0.04)	−0.17 * (−0.33, −0.01)	−0.23 (−0.48, 0.03)
**Basal Ganglia (BG)**	I	Ref	Ref	Ref	Ref
	II	−0.06 * (−0.11, −0.01)	−0.03 (−0.11, 0.04)	−0.05 * (−0.099, 0.004)	−0.04 (−0.11, 0.04)
	III	−0.24 ** (−0.36, −0.12)	−0.14 (−0.33, 0.05)	−0.25 ** (−0.37, −0.13)	−0.16 (−0.35, 0.03)
	IV	−0.42 (−0.85, −0.01)	−0.07 (−0.73, 0.60)	−0.38 (−0.80, 0.05)	−0.07 (−0.73, 0.60)
**Mixed** **Region High PVS Burden**	None	Ref	Ref	Ref	Ref
	Only BG	−0.30 * (−0.48, −0.11)	−0.13 (−0.42, 0.17)	−0.32 ** (−0.50, −0.13)	−0.09 (−0.39, 0.20)
	Only CSO	−0.20 ** (−0.29, −0.10)	−0.17 * (−0.31, −0.03)	−0.18 ** (−0.27, −0.09)	−0.17 * (−0.31, −0.02)
	Both	−0.24 ** (−0.38, −0.11)	−0.17 (−0.39, 0.05)	−0.24 ** (−0.38, −0.10)	−0.21 (−0.43, 0.01)

BG = basal ganglia; CSO = centrum semiovale; FHS = Framingham Heart Study; MRI = magnetic resonance imaging; PVS = perivascular spaces; SE = standard error ^a^ PVS were graded independently in the basal ganglia and centrum semiovale using counts to assign severity: grade I (<10), grade II (10–20), grade III (20–40) and grade IV (>40). High PVS burden was defined as grade III–IV in each region. ^‡^ Model 1 adjusts for age, sex, FHS cohort, and time interval between MRI acquisition and the clinic exam. Model 2 additionally adjusts for diabetes mellitus, hypertension, and smoking status; * *p*-values or q-values < 0.05; ** *p*-values or q-values < 0.001.

**Table 7 brainsci-13-01323-t007:** (**A**) Multivariable analysis of the relation between MRI-visible perivascular spaces by grade and brain region, and both total and regional cortical gray volumes (Model 1 ^‡^). (**B**) Multivariable analysis of the relation between MRI-visible perivascular spaces by grade ^a^ and brain region and both total and regional cortical gray volumes (Model 2 ^‡^).

MRI-Visible Perivascular Spaces (PVS) Grading ^a^		Volumes N = 2452				
		Model 1 ^‡^				
		Cortical Gray β (95% CI)	Cortical Gray Frontal β (95% CI)	Cortical Gray Temporal β (95% CI)	Cortical Gray Parietal β (95% CI)	Cortical Gray Occipital β (95% CI)
**Centrum Semiovale (CSO)**	I	Ref	Ref	Ref	Ref	Ref
	II	−0.06 (−0.12, 0.01)	−0.09 * (−0.15, −0.02)	−0.07 (−0.14, 0.01)	−0.01 (−0.09, 0.06)	0.04 (−0.04, 0.12)
	III	−0.17 * (−0.29, −0.06)	−0.12 * (−0.24, −0.01)	−0.20 * (−0.34, −0.06)	−0.15 * (−0.29, −0.01)	−0.05 (−0.20, 0.10)
	IV	−0.19 (−0.39, 0.00)	−0.23 * (−0.43, −0.03)	0.03 (−0.20, 0.26)	−0.13 (−0.37, 0.10)	−0.23 (−0.48, 0.03)
**Basal** **Ganglia (BG)**	I	Ref	Ref	Ref	Ref	Ref
	II	−0.04 (−0.10, 0.02)	0.01 (−0.05, 0.07)	−0.02 (−0.09, 0.05)	−0.07 * (−0.144, −0.003)	−0.09 * (−0.17, −0.02)
	III	−0.09 (−0.23, 0.06)	−0.01 (−0.16, 0.15)	0.02 (−0.16, 0.19)	−0.21 * (−0.39, −0.03)	−0.14 (−0.33, 0.05)
	IV	−0.43 (−0.95, 0.10)	−0.11 (−0.65, 0.43)	−0.49 (−1.11, 0.13)	−0.24 (−0.87, 0.39)	−0.66 (−1.34, 0.02)
**Mixed** **Region High PVS Burden**	None	Ref	Ref	Ref	Ref	Ref
	Only BG	−0.09 (−0.32, 0.14)	0.03 (−0.21, 0.27)	−0.16 (−0.43, 0.11)	−0.01 (−0.29, 0.26)	−0.21 (−0.51, 0.08)
	Only CSO	−0.15 * (−0.26, −0.04)	−0.09 (−0.21, 0.02)	−0.17 * (−0.30, −0.04)	−0.08 (−0.22, 0.05)	−0.13 (−0.27, 0.02)
	Both	−0.13 (−0.30, 0.04)	−0.08 (−0.25, 0.10)	0.02 (−0.18, 0.22)	−0.28 * (−0.49, −0.08)	−0.11 (−0.33, 0.11)
**(B)**						
**MRI-Visible** **Perivascular Spaces (PVS) Grading ^a^**		**Volumes** **N = 2452**				
		**Model 2 ^‡^**				
		**Total** **Cortical Gray** **β (95% CI)**	**Cortical Gray Frontal** **β (95% CI)**	**Cortical Gray Temporal** **β (95% CI)**	**Cortical Gray Parietal** **β (95% CI)**	**Cortical Gray** **Occipital** **β (95% CI)**
**Centrum Semiovale (CSO)**	I	Ref	Ref	Ref	Ref	Ref
	II	−0.03 (−0.09, 0.03)	−0.06 (−0.126, 0.002)	−0.05 (−0.12, 0.02)	−0.003 (−0.08, 0.07)	0.06 (−0.02, 0.14)
	III	−0.14* (−0.25, −0.02)	−0.10 (−0.22, 0.02)	−0.17 * (−0.3, −0.03)	−0.13 (−0.27, 0.01)	−0.03 (−0.18, 0.13)
	IV	−0.13 (−0.33, 0.07)	−0.16 (−0.36, 0.04)	0.09 (−0.14, 0.33)	−0.14 (−0.38, 0.1)	−0.18 (−0.44, 0.08)
**Basal** **Ganglia (BG)**	I	Ref	Ref	Ref	Ref	Ref
	II	−0.02 (−0.08, 0.03)	0.03 (−0.03, 0.09)	−0.01 (−0.08, 0.06)	−0.06 (−0.13, 0.01)	−0.09 * (−0.17, −0.01)
	III	−0.09 (−0.23, 0.06)	0.005 (−0.15, 0.16)	0.01 (−0.17, 0.18)	−0.21 * (−0.39, −0.03)	−0.15 (−0.35, 0.04)
	IV	−0.38 (−0.90, 0.14)	−0.06 (−0.60, 0.48)	−0.45 (−1.07, 0.16)	−0.21 (−0.84, 0.41)	−0.63 (−1.31, 0.05)
**Mixed** **Region High PVS Burden**	None	Ref	Ref	Ref	Ref	Ref
	Only BG	−0.12 (−0.35, 0.11)	0.02 (−0.21, 0.26)	−0.21 (−0.48, 0.06)	−0.03 (−0.31, 0.24)	−0.24 (−0.54, 0.06)
	Only CSO	−0.12 * (−0.23, −0.01)	−0.07 (−0.18, 0.05)	−0.14 * (−0.27, −0.01)	−0.08 (−0.21, 0.06)	−0.11 (−0.25, 0.04)
	Both	−0.12 (−0.29,0.06)	−0.06 (−0.24, 0.12)	0.04 (−0.17, 0.24)	−0.28 * (−0.48, −0.07)	−0.11 (−0.34, 0.11)

(**A**) BG = basal ganglia; CSO = centrum semiovale; FHS = Framingham Heart Study; MRI = magnetic resonance imaging; PVS = perivascular spaces; SE = standard error; ^a^ PVS were graded independently in the basal ganglia and centrum semiovale using counts to assign severity: grade I (<10), grade II (10–20), grade III (20–40) and grade IV (>40). High PVS burden was defined as grade III–IV in each region.; ^‡^ Model 1 adjusts for age, sex, FHS cohort, and time interval between MRI acquisition and the clinic exam. Model 2 additionally adjusts for diabetes mellitus, hypertension, and smoking status; * *p*-values < 0.05. (**B**) BG = basal ganglia; CSO = centrum semiovale; FHS = Framingham Heart Study; MRI = magnetic resonance imaging; PVS = perivascular spaces; SE = standard error; ^a^ PVS were graded independently in the basal ganglia and centrum semiovale using counts to assign severity: grade I (<10), grade II (10–20), grade III (20–40) and grade IV (>40). High PVS burden was defined as grade III–IV in each region. ^‡^ Model 1 adjusts for age, sex, FHS cohort, and time interval between MRI acquisition and the clinic exam. Model 2 additionally adjusts for diabetes mellitus, hypertension, and smoking status; * *p*-values < 0.05.

## Data Availability

All analysis was performed on data from the Framingham Heart Study. No new data were created in this manuscript.

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
