# Peer review of "Relation of MRI-Visible Perivascular Spaces and Other MRI Markers of Cerebral Small Vessel Disease"

_brainsci, 2023, doi:10.3390/brainsci13091323_

Round 1

Reviewer 1 Report

The submission paper investigated the relationship between MRI perivascular visible spaces (PVS) and the markers of cerebral small vessel disease (CSVD). The paper used the multivariable regression analysis to study the relationships between PVS and CBI, WMH, CMB, and other markers, respectively.

The following comments could be considered for improvement:

1) The discussion section could explain more pathological interpretation on the strong relationships between MRI visible PVS and WMH, and CBI, with some references support to the  discovery.

2) The resolution of Figure 3 should be improved for better visualization.

3) The reference style should follow the MDPI format guidance.

Reviewer 2 Report

brainsci-2571332: “Relation of MRI-Visible Perivascular Spaces and other MRI markers of cerebral small vessel disease”

In this study in humans, the authors are trying to demonstrate a potential of MRI approach in the understanding of an association between cerebral small vessel diseases and perivascular space in the brain. The material is described successively and conclusions are partially supported by obtained data.

Remarks/recommendations:

1.    the list of all abbreviations should be included in the text;

2.    in Figure 3:

a) the plates should be denoted by the letters;

b) the main groups (“Cerebral microbleeds”, “Covert brain infarct”, and “White matter hyperintensities”) should be entitled with additional abbreviations (in brackets);

c) ordinate scales should be unified for each of the main groups;

3.    in line 17, “Perivascular spaces (PVS) visible….”;

4.    in lines 28 and 29, the statement of “PVS are associated with… neurodegeneration markers…” is doubtful in this study;

5.    the sentence in lines 77 and 78 needs to be rewritten;

6.    “Institutional Review Board Statement” and “Informed Consent Statement” (see lines 483-486) should be duplicated in the section of “2. Materials and Methods”;

7.    in line 103, “CSF” should be open;

8.    in lines 287, 296, 305, 313, 322, 331,340, and 349, “<0.001” needs to be clarified as typically two “stars” correspond “<0.01”;

9.    in line 430, “…likely resulting from neurodegeneration,…” needs a reference;

10. in lines 462-466, this final paragraph is absolutely inconclusive;

11. in “References”, up to five authors should be denoted where is appropriate.

 Minor editing of English language required

Round 2

Reviewer 2 Report

brainsci-2571332: “Relation of MRI-Visible Perivascular Spaces and other MRI markers of cerebral small vessel disease”

The authors have made a careful revision and responded to almost all points I raised.

Author Response

We thank the reviewer for their insights and acknowledgement of our improvements on the work.